# Spatiotemporal data analysis with chronological networks

Leonardo N. Ferreira [1,2,3 ✉], Didier A. Vega-Oliveros [4,5], Moshé Cotacallapa[1], Manoel F. Cardoso [6], Marcos G. Quiles[7], Liang Zhao [8] & Elbert E. N. Macau[1,7]

The number of spatiotemporal data sets has increased rapidly in the last years, which demands robust and fast methods to extract information from this kind of data. Here, we propose a network-based model, called Chronnet, for spatiotemporal data analysis. The network construction process consists of dividing a geometric space into grid cells represented by nodes connected chronologically. Strong links in the network represent consecutive recurrent events between cells. The chronnet construction process is fast, making the model suitable to process large data sets. Using artificial and real data sets, we show how chronnets can capture data properties beyond simple statistics, like frequent patterns, spatial changes, outliers, and spatiotemporal clusters. Therefore, we conclude that chronnets represent a robust tool for the analysis of spatiotemporal data sets.

[1] National Institute for Space Research, Associated Laboratory for Computing and Applied Mathematics, São José Dos Campos - SP, Brazil. [2] Department of Physics, Humboldt University, Berlin, Germany. [3] Potsdam Institute for Climate Impact Research, Potsdam, Germany. [4] Institute of Computing, University of Campinas, Campinas - SP, Brazil. [5] Indiana University, School of Informatics, Computing and Engineering, Bloomington, IN, USA. [6] National Institute for Space Research, Center for Earth System Science, São José dos Campos - SP, Brazil. [7] Federal University of São Paulo, Institute of Science and Technology, São José Dos Campos - SP, Brazil. [8] Department of Computing and Mathematics, Faculty of Philosophy, Science, and Letters at Ribeirão Preto (FFCLRP), University of São Paulo, Ribeirão Preto - SP, Brazil. ✉email: ferreira@leonardonascimento.com

Large amounts of spatiotemporal data are collected every day from several domains, including georeferenced climate variables, epidemic outbreaks, crime events, social media, traffic, and transportation dynamics, among many others. Analyzing and mining such kind of data is of great importance for advancing the state-of-the-art in many scientific problems and real applications. Nevertheless, data with spatial and temporal characteristics have different properties in comparison to relational sources studied in classical data mining literature[1]: they present temporal and/or spatial dependencies, in which instances are not independent or identically distributed. It means that samples can be structurally related in some spatial regions or specific temporal moments. Also, they are non-static, i.e., the instances can change their class attribute depending on time and location. Thus, traditional data mining methods are not the ideal tools for spatiotemporal data, which can result in poor performance and misleading interpretation[1,2].

Spatiotemporal data mining (STDM) is an emerging field that proposes novel methods for analyzing event-based data, trajectories, time-series, spatial maps, and raster data[1]. Some of the main STDM tasks are clustering, anomaly detection, frequent pattern mining, relationship mining, change detection, and predictive learning. STDM brought new challenges and opportunities in several application domains. In this context, some works have employed networks as a way of representing and analyzing spatiotemporal information[3–6]. The interest is justified by the benefits that the network representation provides, like the ability to describe sub-manifold in dimensional space and to capture dynamic and topological structures—hierarchical structures (communities) and global or local patterns, independent of the data distribution[6,7]. The network-based representation of spatiotemporal data also allows the discovery of different patterns like local-range spatial dependencies and long-distant correlations[4,5,8].

Functional or correlation networks[9] have been widely applied to map spatiotemporal data into networks. Such a method connects nodes, i.e., time-series from spatial grid-cells, according to their correlation. Functional networks have been employed in a diversity of areas like: in earth sciences[4,5,8,10,11], bioinformatics[12], finance[13], among others. One drawback of this kind of networks appears when short-length time series are considered, making the statistical significance of correlations questionable and may result in spurious links in the network. Additionally, in the case of event-based data, it is common to have a higher proportion of—no events—or values equal to zero throughout time, which clearly affect the correlation results. Alternative methods have been proposed to construct networks[6,7,14], like the visibility property in time series[14]. However, this method produces single and static representations of the time-series, discarding the spatial location.

Another way to construct networks considers the chronological order in the spatiotemporal events to define the links. It was applied to study seismic events by constructing a network from a geographic region divided into small cubic cells[15,16]. It also was used to analyze wildfire events in the Amazon basin by constructing the network following the co-occurrence of events in a grid division of the area[3]. The authors presented two approaches—a single-layer network and temporal network—for data analysis. Although the previous works have identified valuable information in a couple of domains, the proposed methods and results are restricted to the specific problem and were not generalized to a variety of domains with spatiotemporal events.

Here, we provide a general method for representing spatiotemporal events with networks, as a structure for dealing with data mining problems, like clustering, predictive learning, pattern mining, anomaly detection, change detection, and relationship mining. The idea is to divide the geometric space into grid cells that are represented by nodes. Links are established between the nodes in chronological order, i.e., nodes are connected if successive events occur in the respective cells. Therefore, the links represent events in a chronological occurrence, which we call, for the sake of simplicity, as chronnet. In this way, we generalize the previously reported mechanism of connections[3,15,16] transforming the spatiotemporal data set into a network, as explored in other domains like text-mining[17,18] and social media[19,20].

In such a way, recurrent consecutive events are represented by strong (high weighted) links. The constructed network presents non-trivial characteristics obeying the original spatiotemporal events distribution, such as scale-free and small-world for power-law event distribution. The chronnets allow the detection of a new type of spatiotemporal community and permit the identification of influential nodes and outliers. More importantly, the method is computationally efficient—with linear complexity, can be easily parallelized and distributed, and can be adapted according to the domain, e.g., different number of sequence connections, grid division, and time windows in the case of temporal analyses.

The network construction method has been tested and analyzed using artificial and real data. We propose a simple data set generator to explain how the topology of the chronnet captures different spatial and temporal characteristics. Specifically, we use the proposed data set generator to experimentally describe how chronnet measures can be used to extract information from spatiotemporal data. We also applied our model to a real data set composed of global fire events as a case study. The results show that our model is able to describe the frequency of fire events, regions with outlier activity, and the fire seasonal activity using a single chronnet and complex network measures.

## Results

**Artificial data sets and dynamical systems.** In this section, we construct data sets to generate static chronnets, i.e., the whole data set is used to construct a single chronnet. We employ this single network to characterize the whole historical activity using network theory. We provide interpretations for the network measures in the context of chronnets and show how these measures can be used to extract information from spatiotemporal data.

Figure 1 presents three spatiotemporal data sets constructed using our data generator. Each data set was constructed considering a different probability distribution (P): uniform, power-law, and exponential. For each of them, we also present the undirected chronnets constructed for and their degree distributions. The uniform probability distribution generates a Gaussian degree distribution, the power-law distribution creates a scale-free[21] chronnet, and the exponential probability distribution generates a chronnet with exponential degree distribution. Scale-free networks are characterized by the presence of many low degree nodes and just a few high degree nodes (hubs). Exponential degree distributions have a fast decay, which makes the hubs less numerous than what was expected in a scale-free network. The presence of hubs reveals interesting features that can be explored. In summary, the degree distribution of a chronnet captures the probability distribution of event generation.

Figure 2 illustrates an artificial data set generated with a probability distribution P following a power-law. In this case, the maximum probability $p \in P$ and the total time of generation T are higher than the previous case (Fig. 1d). When P or/and T are high enough, the resulting data set have a large number of events and the resulting chronnet tends to be highly connected ($\langle k \rangle \approx n - 1$). In these situations, the node strength provides more information than the node degree. In Fig. 2b, we show that the strength

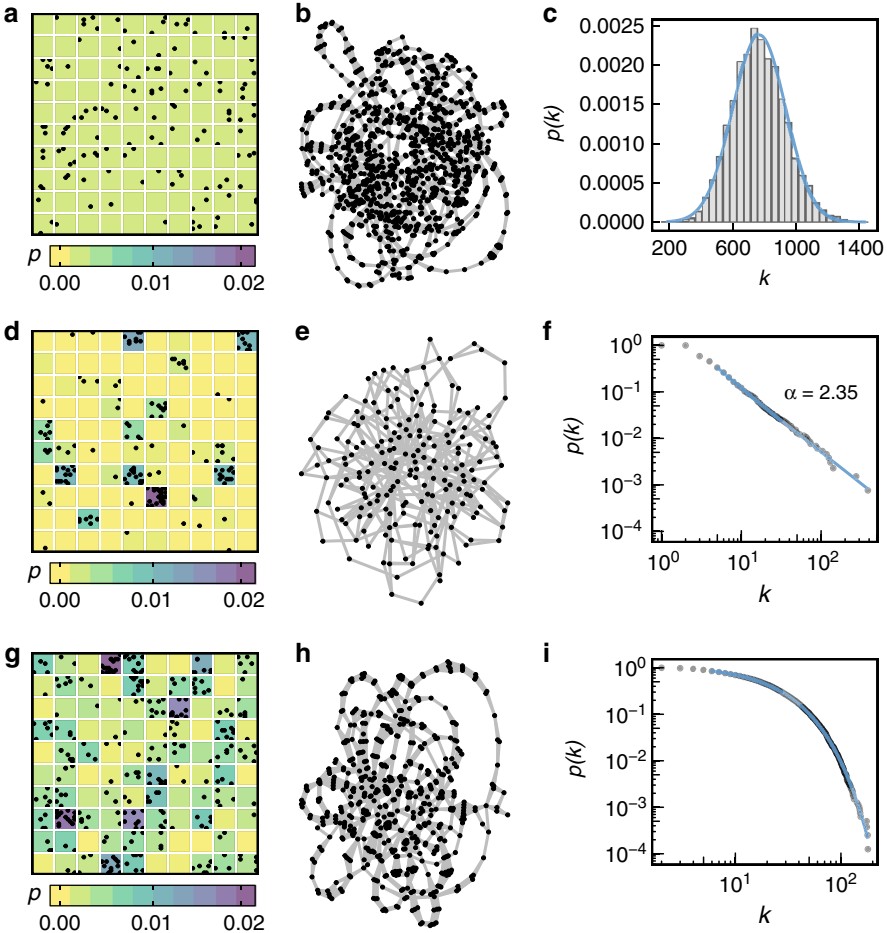

**Fig. 1 Chronnets constructed with different spatiotemporal data sets generated with three probability distributions. a** uniform, **d** power-law, and **g** exponential. For simplicity purposes, we plot part of the data set. For each distribution, we illustrate the respective networks (**b**, **e**, and **h**) and degree distributions (**c**, **f**, and **i**).

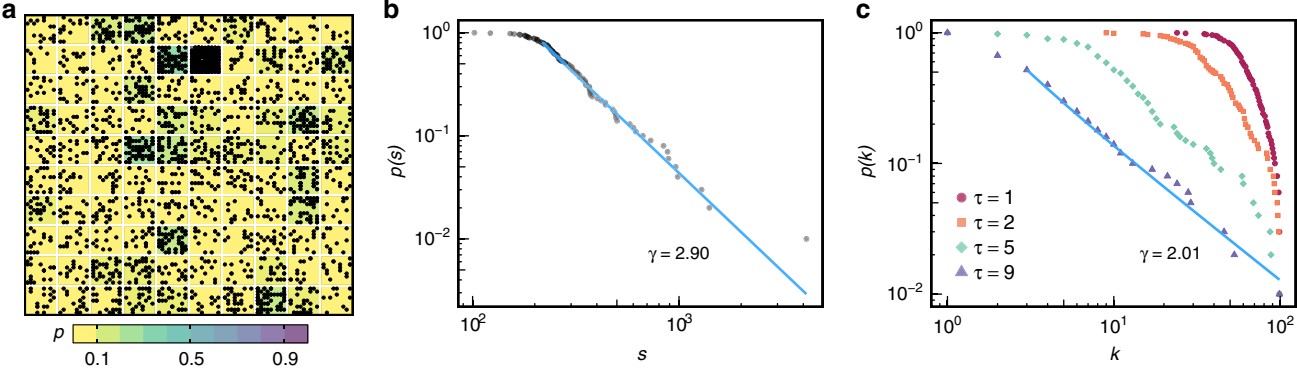

**Fig. 2 Retrieving the probability of event generation in densely connected chronnets. a** An example of a data set with a power-law distribution of event generation that creates a highly connected network. In these cases, the strength distribution of the resulting chronnets **b** provides a better description of the network. Another possibility is to consider a pruning threshold ($\tau$) to remove weak links and to analyze the degree distribution. **c** Shows the degree distributions for $\tau = 1, 2, 5,$ and $9$ represented by different colors and marks. The blue lines (**b** and **c**) represent the power-law fitting. In both cases (**b** and **c**), it is possible to observe the original power-law distribution from the original data set (**a**).

distribution can capture the power-law probability $P$. From the strength distribution it is possible to observe that chronnets have weak nodes. In Fig 2c, we show the influence of the weak links pruning on the topology of the chronnet considering the same artificial data set from Fig. 2. As previously described, the resulting chronnet for this data set is densely connected, which does not bring much information. However, it is possible to

reveal the original probability distribution $P$ by pruning weak links from the network ($\tau = 9$). The pruned chronnet has a degree distribution that follows a power-law ($\gamma = 2.01$), similar to the probability distribution ($P$) used to construct the data set.

The pruning process has a strong influence on the network. The edge density ($D$) presents a fast decay event with small values of $\tau$. For example, in Fig. 2 when $\tau = 9$, only 6% of the original

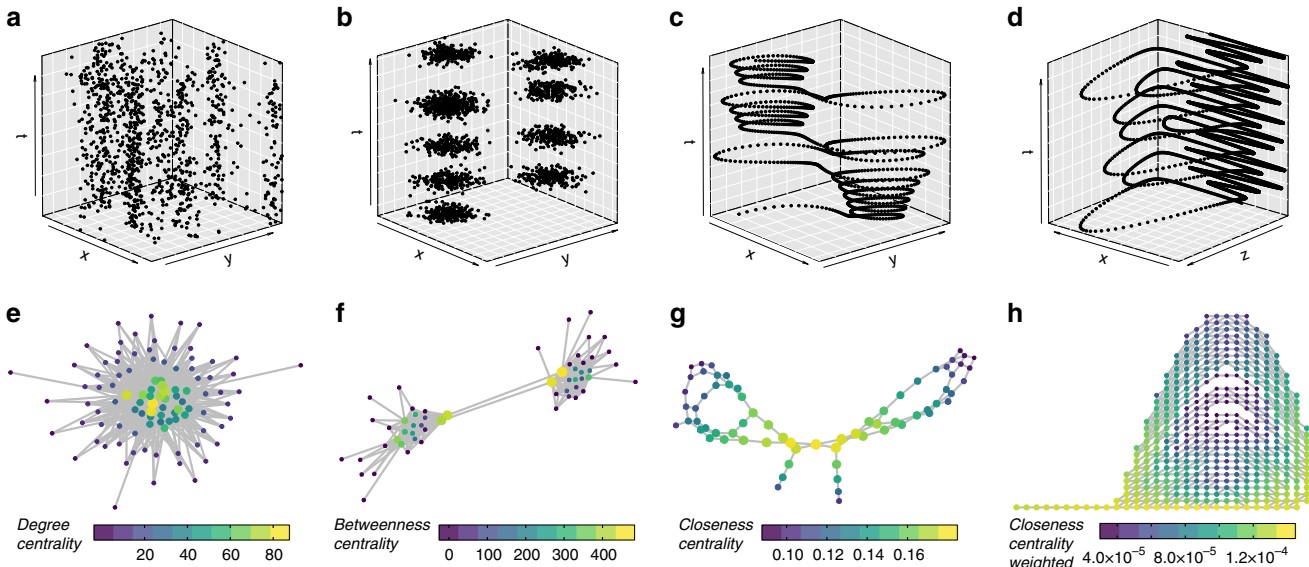

**Fig. 3 Node centrality in chronnets.** The upper three figures **a**–**d** illustrate data sets used to construct the chronnets, respectively illustrated below them **e**–**h**. Only parts of the data set are shown for illustrative reasons. **a** Data set constructed considering the power-law probability illustrated in Fig. 1d and $T = 10^5$. **b** Data set constructed by generating 40 Gaussians ($\sigma = 100$) that alternate in two different locations ($T = 10^6$). **c** Data set constructed by sampling ($T = 200$ and $\Delta t = 0.01$) the $x$ and $y$ coordinates of one realization of the Lorenz system[35] with parameters $\sigma = 10$, $\beta = 8/3$, and $\rho = 28$, resulting in a chaotic trajectory. **d** Data set constructed by sampling ($T = 1000$ and $\Delta t = 0.02$) the $x$ and $z$ coordinates of one realization of the Rössler system[35] with parameters $a = 0.2$, $b = 0.2$, and $c = 5.7$, also resulting in a chaotic trajectory. The chronnets were constructed considering grid sizes **e** and **f** $Gr_{10\times10}$, **g** $Gr_{15\times15}$, and **h** $Gr_{30\times30}$. Weak links were pruned from all the chronnets except for the Rössler system **h** considering **e** and **f** $\tau = 1$ and **g** $\tau = 15$. The node size and colors represent the respective centrality measure in the legend below each network.

links remain. This fast decay reflects the strength distribution that shows that the great majority of nodes have low strength. It means that even a very low threshold will remove many links. Therefore, the pruning threshold $\tau$ cannot be high otherwise it will remove too many links and might make the network disconnected. When the pruning threshold is low, it is possible to remove weak links but still preserve part of the strongest links and the nodes in the largest component. These results show, for a specific data set, how the pruning affects the chronnet and how it can be used to reveal interesting topological features. Other data sets may respond differently to the pruning process, which means that the pruning should be analyzed in particular.

Important nodes in chronnets can be detected by applying network centrality measures[22]. In Fig. 3, we present four data sets with some specific temporal characteristics. The first data set was constructed using the power-law probability distribution illustrated in Fig. 1d. The resulting chronnet (Fig. 3e) shows that the cells with higher event generation probability ($p$) tend to generate a high number of connections. These hubs can be captured by the degree centrality. The second data set (Fig. 3b) was constructed by generating Gaussian distributions in two fixed positions ($x$ and $y$) alternated in time, forming spatiotemporal clusters. The resulting chronnet (Fig. 3f) is composed of two communities. By definition, communities have high intra-community links but low inter-communities connections. This feature makes the betweenness higher in links and nodes that connect communities. Therefore, the betweenness centrality can be used to detect nodes connecting communities. The third and fourth data sets (Fig. 3c, d) were constructed by sampling two coordinates of the Lorenz and Rössler systems, respectively, in chaotic regimes. The resulting chronnet for the Lorenz equations (Fig. 3g) has a topology that reproduces the characteristic two-loop form of the trajectory connected by a single node. This node has the shortest average distance between the other ones, which indicates that the represented cell has the shortest temporal distance to all the other

ones when considering all the combinations of cells. This central node has the highest closeness centrality. The chronnet for the Rössler system (Fig. 3h) also reproduces in the topology some characteristics of the trajectory. Most part of the time, it oscillates in the $x$-dimension leading to stronger links between the nodes that represent these cells. Nodes with stronger links have higher closeness centrality when the weights are considered[23], a measure that can be used to find these nodes.

In Fig. 4, we illustrate the community detection process. First, we generate a spatiotemporal data set composed of four different periods. Different event probability distributions ($P$) were arbitrarily chosen in such a way that creates four temporal clusters (Fig. 4b) but also generates outlier events with a smaller probability. We constructed the chronnet and applied the Fast greedy algorithm[24] to find hierarchical community structures (Fig. 4c). The best partition (best modularity) for the chronnet is formed by four clusters (Fig. 4d) which exactly correspond to those ones whose events occur in the same four periods in $T$ during the data set construction. By choosing the number of communities (dendrogram), it is possible to use chronnets to cluster spatiotemporal data set in distinct spatial regions. For example, the same chronnet can be divided into two communities by cut the dendrogram into two clusters. This partition splits the time $T$ into two intervals that correspond to the first and the last two periods.

We applied our spatiotemporal data clustering method to the artificial data set presented in Fig. 4 and compared to the clustering result for the ST-DBSCAN, a well-known clustering algorithm for spatiotemporal data[25]. For the ST-DBSCAN, we tested all the combinations for the three parameters: $eps = \{0, 10, 20, …, 3000\}$, $eps2 = \{0, 10, 20, …, 3000\}$, and $minpts = \{0, 5, 10, …, 200\}$. Our results show that the proposed method can correctly cluster all the points while ST-DBSCAN achieves a maximum adjusted rand index of 0.94. As illustrated in Fig. 4e, the ST-DBSCAN algorithm mainly fails to cluster the outlier

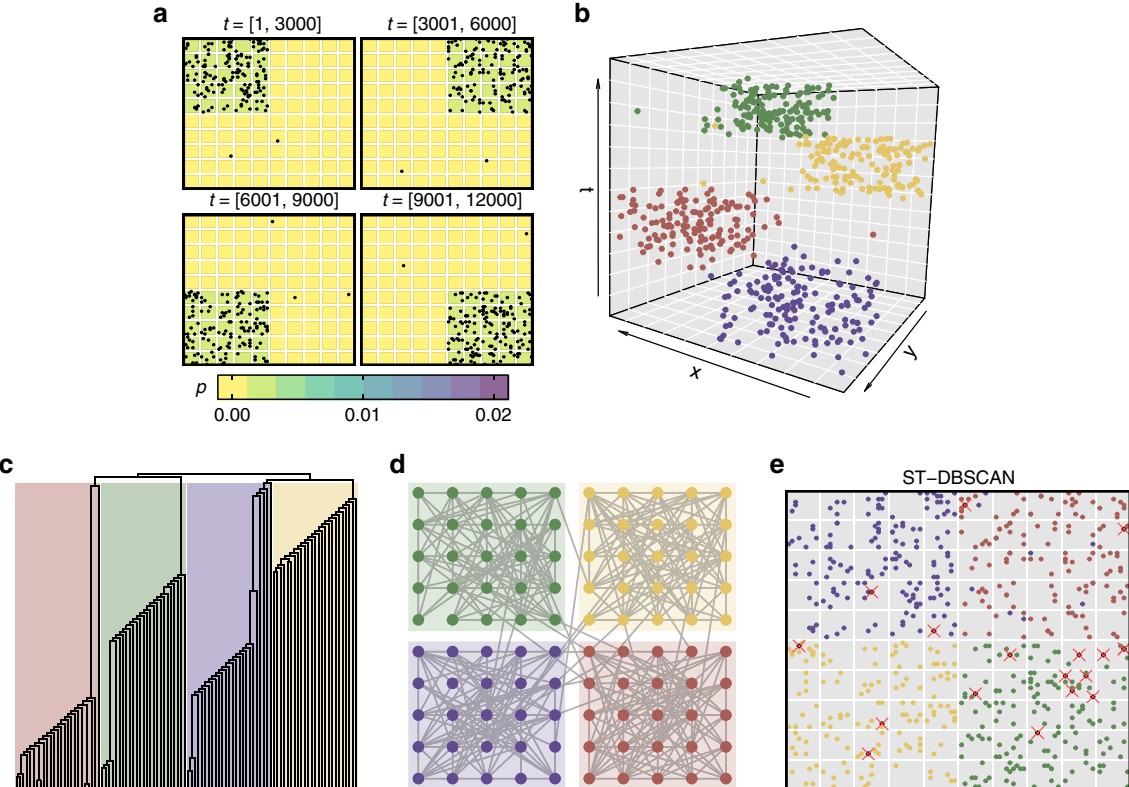

**Fig. 4 Community structure in chronnets. a** We propose an artificial data set constructed by changing the event probability distribution $P$ in four intervals with total time $T = 12,000$. Each of the four small grids illustrates the probabilities and the generated events in those intervals. A few outlier events were generated, represented by points outside the region with higher probability. **b** The resulting data set is composed by merging the events generated during the four periods, represented by colors. **c** Dendrogram obtained by the Fast greedy algorithm[24]. Different colors represent the four communities achieved with the best modularity division. **d** The resulting chronnet is presented in a grid layout that corresponds to the same position in the data set grid. **e** Spatiotemporal clustering using ST-DBSCAN[25] where "×" denotes events that were incorrectly clustered.

events. This result shows that the proposed can successfully find temporal clusters.

Spatial changes can also be detected using the community structure in chronnet. As described in the "Methods" section, change points are defined as changes in the communities where the events occur in time. In Fig. 5 we illustrate this process using an artificial data set composed by groups of points forming spatiotemporal Gaussians that alternate in time. The resulting chronnet (Fig. 5b) is composed of three communities that represent the three spatial regions where the events occur. For every event in the data set, we construct a time series (Fig. 5c) with the node community where each event is located. Step changes in this time series represent spatial changes in the data set.

**Real case application using global fire detections data**. Here, we apply chronnets to analyze a real spatiotemporal data set composed by fire detections. Similar to the previous analysis with the artificial data sets, we use a single chronnet and network measures to describe the historical fire activity (2003–2018). To remove some weak links, we considered a small pruning threshold $\tau = 2$ which keeps 32% of the original links.

In Fig. 6, we present the degree and strength distributions for the resulting chronnet. High degree nodes account for cells whose fire activity occurs consecutively to other cells while the strength captures the number of events. The strength does not capture here the total number of events since links were pruned. In general, regions with high degrees tend to have high strengths.

Both measures show the regions with higher fire activity, e.g., Brazil, large parts of Africa, China, and North Australia. This pattern was observed in previous studies[26,27].

The degree distribution follows a power-law ($\gamma = 4.28$) after a low-degree saturation. Power-law degree distributions are characterized by the presence of many low degree nodes and a few high degree nodes (hubs). The degree exponents $\gamma > 3$ show that the distributions decay fast to make the hubs less numerous than what was expected in scale-free networks[21]. Therefore, we cannot affirm that the network is scale-free, but the presence of hubs reveals interesting features that can be explored. In the fire activity data set, hubs represent cells with periods of fire activity much longer than the other ones. In some cases, these uncommonly long activities are accounted for false wildfire detections like in regions with hot bare soils[28]. In other cases, hubs represent volcanoes or gas flares from oil and gas exploration. The outlier cells found by our method correspond to the cells with the highest number of fire detections marked as outliers in the original data set. This result is expected and shows the accuracy of the method.

The strength distribution can be partially fitted as a log-normal ($\mu = 7.21$ and $\sigma = 1.79$) and a power-law ($\gamma = 4.20$) tail. The distribution shows that there is a high number of nodes with low strength ($s \leq 10^2$), which mainly represent cells in medium and high latitudes (e.g. Patagonia region and the Nordics) with very low fire activity. After this low-strength saturation, the number of nodes decreases as a log-normal distribution until a transition point ($s \geq 5.1 \times 10^4$), where the number of nodes decays faster as the fitted log-normal distribution, which is better represented as a

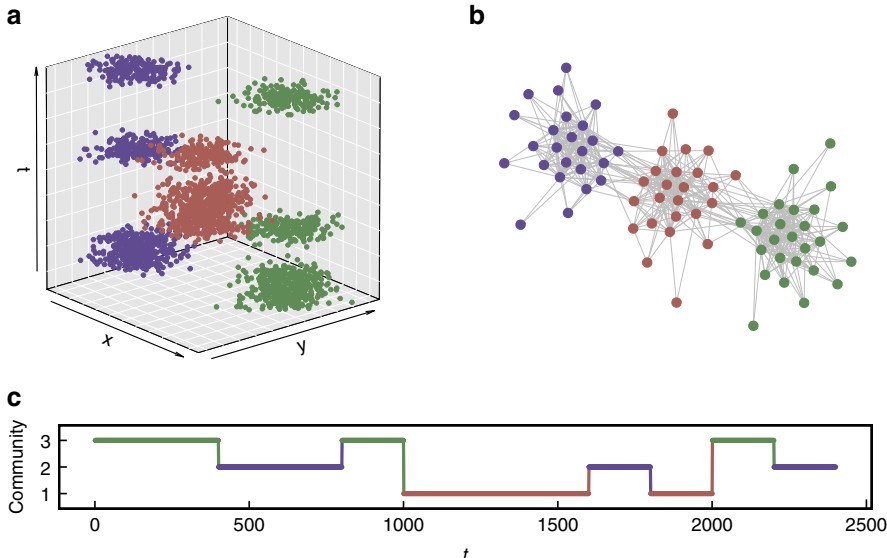

**Fig. 5 Detecting spatial changes with chronnets. a** Artificial data set composed by groups of points sampled from Gaussian distributions in three spatial locations (point color) alternating in time. **b** The resulting chronnet is composed of three communities (node colors) obtained by the Fast greedy community detection[24]. **c** A time series with the node community where each event from the data set.

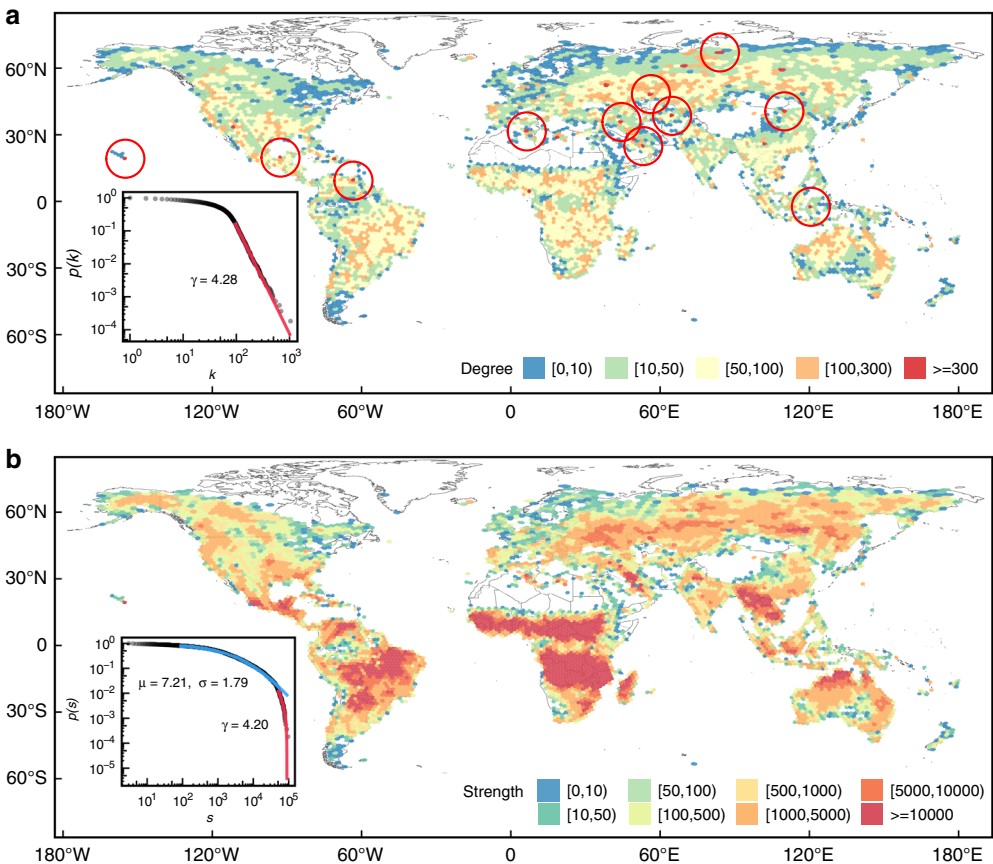

**Fig. 6 The degree and strength distributions for the chronnet constructed with the wildfire data set. a** Degree and **b** strength distributions. A small pruning threshold $\tau = 2$ was considered. The red circles (**a**) mark the 2% of higher degree nodes. The inner plots show the probability distributions. Red and blue lines illustrate the fit of power-law and log-normal distribution, respectively[39].

power-law. This interval comprises the regions with medium ($10^3 \le s \le 10^4$) like the Russia and USA, and high strength nodes ($s > 10^4$), which correspond to high fire activity regions occurring mainly in the tropics, like Brazil, Central America, north Australia, Indochinese Peninsula, and large parts of Sub-Saharan Africa. This later is where the highest strength nodes appear (power-law tail), corresponding to the region in the southern Democratic Republic of the Congo and Northern

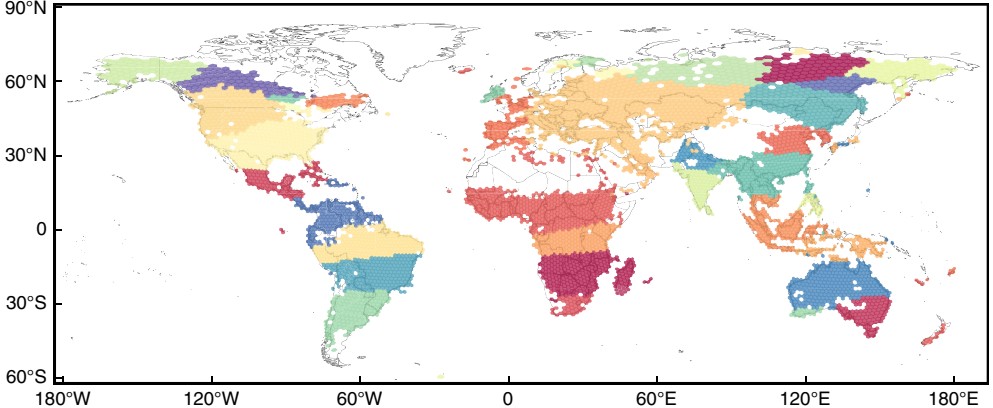

**Fig. 7 Communities in the pruned chronnet constructed with the wildfire data set.** Only 20% of strongest links were considered and 1% of the largest degree nodes were removed (outliers). Communities were achieved using the label propagation method[40]. Colors represent the 38 communities.

Angola, as well as Southern Sudan and the Central African Republic, already described in previous works[26,27].

As demonstrated, simple chronnet measures like the degree and strength can be used to describe the distribution of the events in the data set. Other network measures can be used to describe different aspects of the data set. The average path length $L = 3.21$, indicates that the average time steps to occur consecutive events between any pair of cells is on average small. The transitivity $C = 0.34$ shows that there is a clustering structure in the network, which is an expected result since close regions tend to have similar climate and land use. Since $L$ is small but $C$ is not, the resulting chronnet presents the small-world feature[21,29,30]. The presence of hubs is highly responsible for decreasing the average path length. It means that, for any pair of nodes on the globe, the average number of edges, i.e., consecutive events, to co-occur wildfires is low.

Another capability of chronnets is to represent groups of cells whose activity occurs in the same period as communities. Figure 7 illustrates community structures for the wildfire data set. Most of the communities are formed by nodes that represent relatively near geographical regions. The spatial distribution of the communities can be explained by the similarity of climate and land use in near regions that tends to have fire seasons in the same period[27]. However, some communities are formed by distant regions. For example, the southwest-most community in South America (light green) is the same one in the small region in South Western Australia. It is important to note that we have presented the results for just one network partition (38 communities). Other algorithms, as the fast greedy algorithm[24], returns a hierarchy of communities that can be used to divide the area of study in an arbitrary number of regions. In the fire data set, this hierarchy can be used to study the fire seasons in different geographical scales.

## Discussion

In this paper, we present a network-based representation model for spatiotemporal data analyses called chronnets, whose main goal is to capture recurrent consecutive events in a data set. In this way, we transform spatiotemporal events into a network with a construction method that is fast (linear time complexity) and thus can be applied to large data sets. The model permits the use of different tools from network science and graph mining to extract information from spatiotemporal data sets. It can be used not just to describe, but also to compare data sets.

Our results show that the method can extract simple statistics as well as intricate and hidden information. Local network measures, as the degree and strength, can describe the event activities in each cell and can be used to detect cells with outlier activities, while centrality measures can be applied to detect important nodes. Global network measures describe the data set as a whole and can reveal features that cannot be detected by simply looking to single cells. For example, high transitivity indicates a high tendency of consecutive events to form triangles and cluster together. In these cases, community detection algorithms can be used to find communities, which represent groups of cells whose activity occurs in the same period of time. We also show how to apply our model to some STDM tasks, like frequent patterns detection, spatial changes identification, clustering, and outlier detection.

The aforementioned capabilities of the model were experimentally demonstrated using toy data sets generated by a simple probabilistic process here proposed. The model was also applied to real data set of global active fire detections, in which we described the frequency of fire events, outlier fire detections, and the seasonal activity, using a single chronnet. Some interesting aspects were found on the real data set. The average path length, which indicates that the average time steps to occur consecutive events between any pair of cells, is on average small. The chronnet presents a clustering structure with larger transitivity values, which is an expected result since close regions tend to have similar climate and land use. Then, the chronnet has the small-world property. The larger presence of hubs is responsible for decreasing the average path length. This result indicates that for any pair of nodes on the globe, the average number of edges, i.e., consecutive events, to co-occur wildfires is low. Therefore, we conclude that chronnets are a robust and fast model for spatiotemporal data analysis.

This work is extendable in many directions. The construction method can be improved by introducing controlling parameters to capture not just consecutive events but different lags. It will permit the detection of temporal patterns in other time scales. More variables can also be considered by applying the same grid division process used to transform the spatial coordinates into nodes. Another future work is the chronnets analysis from a temporal network perspective. Different methods of graph mining can also be used with chronnets to propose new pattern recognition techniques. The temporal patterns captured by chronnets can be used to propose new machine-learning tools or forecasting methods. In the particular application problem, new analyses focusing on some specific regions with high fire intensity[27], such as the Amazon forest, South America, or Africa, can be developed. Furthermore, other spatiotemporal data sets may also be analyzed with chronnets.

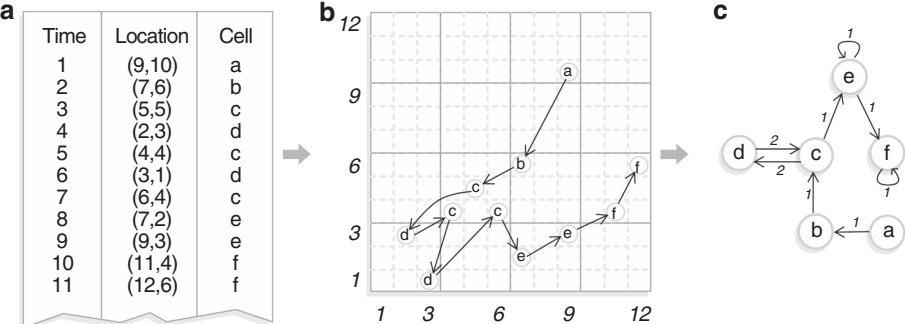

**Fig. 8 Chronnet construction method. a** Given a time-ordered spatiotemporal data set, the first step consists of dividing the area of study into grid cells. **b** Shows a spatial representation of the data set using links between cells to indicate the temporal order they occur, in a sliding window of $h = 1$. **c** In a chronnet, nodes represent grid cells and links stand for co-occurred events between cells. Link weights indicate the number of consecutive events observed between cells.

## Methods

**Chronnet construction.** Our method aims to transform a spatiotemporal data set into a network whose links represent events in a chronological approach, simply named chronnet. Let us consider a spatiotemporal data set $X$ composed of time-ordered sequence of events $X = \{e_1, e_2, \ldots, e_T\}$, where an event $e_a = (l_a, t_a)$ is represented by its location $l_a = (x_a, y_a)$ and time $t_a \in \{t_1, t_2 \ldots \le t_T\}$. Note that an event may also provide more measurements as additional information about the occurrence. The method for constructing the chronnet $G(V, E)$ from the data set consists of three steps. In the first step, the studied geometric area is divided into grid cells and each grid-cell is represented by a node $v_i \in V$. The second step consists of defining a time length for the network construction. The time length $\Delta t$ divides the spatiotemporal data set into time windows. If $\Delta t = T$, the whole data set will be used to generate the single network. If $\Delta t < T$, then each time window will be used to create a layer (snapshot) of the temporal network. The final step comprises the network connections.

Formally, a directed link $(v_i, v_j)$ is established from node $v_i$ to node $v_j$ if two consecutive events $e_a$ and $e_{a+h}$, within a time window of a predetermined size $h \ge 1$, co-occur therein the cells $i$ and $j$, respectively. A threshold parameter $d_{max}$ may be considered to limit the maximum distance between links. In this case, $v_i$ and $v_j$ are connected iff the distance between two consecutive events $e_a$ and $e_{a+h}$ is smaller than $d_{max}$, i.e., $\exists (v_i, v_j) \iff d(e_a, e_{a+h}) \le d_{max}$. The threshold parameter $d_{max}$ can be used to avoid that events occurring in far regions be connected.

The process slides over the entire data set from the first to the last event. The link weight $w_{ij}$ between two nodes $v_i$ and $v_j$ is the sum of consecutive events between them. Note that $i = j$ represents self-loops used to count consecutive events that occur in the same cell. The main idea behind this construction process is to map recurrent consecutive events into strong (high weighted) links. The steps to construct a chronnet with a sliding window $h = 1$ is illustrated in Fig. 8.

In the following, we summarize the chronnet construction process. The first step consists of the grid division where the spatiotemporal data is separated into grid cells. Each cell is represented by a node in a chronnet. The second step corresponds to the definition of time window length. The whole period from the data set can be used to construct a single static network or the data set can be divided into time windows used to create layers (or snapshots) of a temporal network. Finally, the node connections are established by adding directed links between two nodes if two events co-occur within a sliding window of size $h$ in the cells represented by them. Self-loops are used to denote consecutive events in the same cell.

Some comments on this construction method are remarked. First, it contains one parameter, the sliding window $h$, and one domain decision, the grid division which is a scale of coarse-graining. Since there is no priori rule to determine the grid size, it is a decision related to the application domain. Increasing the resolution impacts in the final network size and sparsity. On the other hand, lowing the resolution helps to remove redundancies in the observations, but may also result in a loss of spatial information. The same happens with the parameter $h$. The larger $h$, the more events are temporal aggregated and the denser the network. Both sliding window and grid size are primary decisions independent of performing temporal or single-network analysis.

Chronnets are by definition directed weighted graphs. In this paper, for simplicity purposes and without loss of generality, we opt for focusing on analyzing undirected weighted chronnets constructed by consecutive events ($h = 1$) without distance restriction ($d_{max} = \infty$). In an undirected chronnet, link weights represent the total number of events that co-occurred between two cells independently of which node appears first in time, i.e., the sum of in-link and out-link weights. In the rest of this paper, we use the term "chronnet" to refer to an undirected weighted network.

We consider in this paper, for simplicity reason, that the data set does not have parallel (same timestamp) events. However, this is not a limitation in the model. Parallel events can be treated by establishing links between all the combinations of

different nodes with events in consecutive time stamps. Considering a time window $h = 1$ and two sets of nodes $\{v_1, v_2, \ldots, v_r\}^t$ and $\{v_1, v_2, \ldots, v_s\}^{t+1}$ whose cells have parallel events in time $t$ and $t + 1$, respectively, a link is established between all the combinations of different nodes between both sets. We also consider only the spatial ($x$ and $y$) locations and time of the events in the construction method but we emphasize that our proposed approach is not limited to these three variables. Our approach can be extended to more variables by applying the same grid division process to other variables.

Weak links in chronnets represent sporadic consecutive events between two cells that might not represent temporal patterns. These links may appear as a result of different factors. One common reason is the inaccurate or wrong records generated by distinct problems in the data gathering process[31], leading to spurious links in the chronnet. To minimize these problems, we consider an optional pruning step that consists of removing links whose weights are equal or lower than a threshold $\tau$. Pruning weak links will remove some of these wrong records from the analyses and can be used as an outlier removal step. This process can also be used to transform weighted chronnets into unweighted ones. In this case, only those links whose weights are higher than $\tau$ remain in the network.

An advantage of the construction process is its low time complexity. The time complexity for finding the cells whose events occur (grid division) is linear concerning the number of events if a squared grid is considered. Assuming no parallel events and that the events were recorded in chronological order (sorted) in the data set, the time complexity for constructing the chronnet is also linear to the total number of events $T$. If the data set is unsorted, the time complexity is bounded by the time complexity of the sorting algorithm. Even in the unsorted case, the time complexity is low, considering the fast sorting algorithms available. This process can be easily parallelized by breaking the sorted data set into smaller data sets, constructing smaller chronnets, and merging them. The parallelization can considerably speed up the construction process. Therefore, our method can is suitable to model large-scale spatiotemporal data sets.

**Chronnet characterization.** The degree $k_i$ of a node $v_i$ in a chronnet represents the number of other cells whose events occur consecutively in time with respect to $v_i$. Higher degree nodes (hubs) represent cells with relatively longer periods of activity occurring simultaneously with many other cells. The activity period in a hub cell should be sufficiently higher than the others to allow the connection to several other cells.

The strength $s_i = \sum_j^n w_{ij}$ is the sum of the weights of the node $v_i$ and represents the frequency that the events happened after or before any neighbor. A high node strength indicates a recurrent link between this node and at least one of its neighbors. High strength accounts for a higher number of events.

The degree distribution is the fraction of nodes with the same degree $k$, which could be seen as the probability $P(k)$ of randomly selecting a node with degree $k$[21]. Recalling that in chronnets an edge represents consecutive events between two cells, the degree distribution represents how spatially distributed are the events over time. The more uniform is the degree distribution, the more spatially homogeneous occurred the events. When the chronnet exhibits heavy-tailed degree distribution, e.g. following a power law in the form $P(k) \sim k^{-\gamma}$, then, most of the cells have a low degree, i.e., they present few events that co-occurred with other nodes. On the other hand, few cells have very high degrees (hubs), which means, the nodes present a larger number of spatiotemporal events that co-occurred with many other nodes of the grid. The strength distribution interpretation is analogous to degree distribution. In this case, the frequency of consecutive events (link weights) between nodes is also taken into account.

A path length in chronnets represents the number of consecutive events in a route between two nodes, i.e., how many consecutive historical events separate two cells. The simplest case is the one-path distance between nodes $v_i$ and $v_j$, meaning that both nodes were consecutively activated after a while. A two-path distance

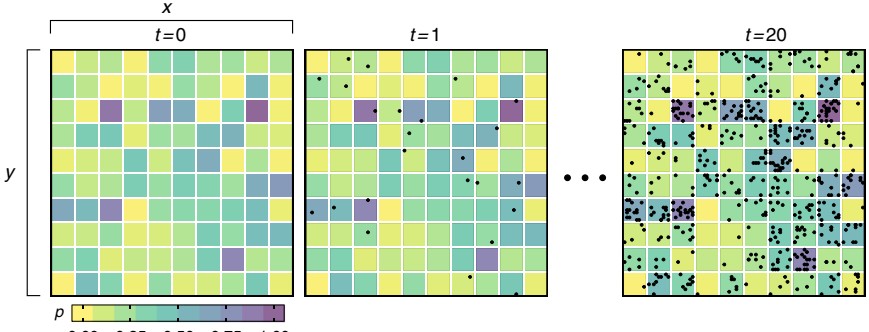

**Fig. 9 Spatiotemporal event generator.** An example of a data set constructed considering a grid $Gr_{10 \times 10}$, an exponential probability distribution (cells' colors), and a time length $T = 20$. Each grid represents all the generated events (black dots) until that time.

means that it was necessary an intermediate or third node $v_k$ to occur a consecutive cascade of events between $v_i$ and $v_j$, and so on for larger distances paths. Therefore, the shortest path distance, $\ell_{i,j}$, is the minimum sequence of historical cascade activation that separates two nodes.

The average shortest path ($\langle \ell \rangle$) corresponds to the expected shortest path distance to occur an event activation between any pair of nodes. Then, the diameter ($\max(\ell_{i,j})$) is the longest shortest path distances to historical co-occur a cascade of events between the nodes on the chronnet.

Centrality measures quantify how influential or central nodes are, which can be defined in many ways[22]. It is important to mention that many centrality measures do not take into account the link weights. These measures are inappropriated for chronnets since they consider that strong (recurrent events) and weak links have the same influence. Therefore, two approaches can be considered: to use centrality measures for weighed graphs[23] or to prune low weight links and then apply centrality measures for unweighted networks[22].

One of the most straightforward definitions, the degree centrality, assumes that the most relevant nodes are those with the highest degrees. Considering a node $v_i$ and its degree $k_i$, the degree centrality is $C_i^{\mathrm{deg}} = k_i$. As discussed previously, high degree nodes in chronnets represent cells whose events occur consecutively to other many other cells. Another common measure is the betweenness centrality that quantifies how many times a node is traversed when considering the shortest paths between all node combinations in the network, defined as $C_i^{\mathrm{betw}} = \sum_{jk} n_i^{jk}/g^{jk}$, where $n_i^{jk}$ is the number of shortest paths between nodes $v_j$ and $v_k$ that passes through $v_i$ and $g^{jk}$ is the total number of shortest paths between $v_j$ and $v_k$. The betweenness centrality can detect nodes connecting communities (densely connected nodes with few connections between groups). The closeness centrality, defined as the inverse of the sum of distances $d_{ij}$ between all combination of node $v_i$ and $v_j$, i.e., $C_i^{\mathrm{clo}} = 1/\sum_{j=1}^{n} \ell_{ij}$. In chronnets, the closeness centrality can be used to find those nodes whose consecutive events occur with less time steps to all the other ones. Some centrality measures were also extended to weighted graphs[23]. The closeness centrality, for example, takes the weights into account when calculating the $\ell_{ij}$ between nodes.

Transitivity accounts for the proportion of cycles of order three on the network, which is a common property found in real-world networks[21]. In chronnets, a triangle means a recurrent or reinforcement pattern of cycle activation between three nodes. If the events occur at random positions, we have a very low transitivity coefficient in the chronnet. In the case of weighted and directed networks, it can represent strong causal or reinforcement behavior, e.g., a modus operandi, or seasonal fire progression[3].

A common pattern in spatiotemporal data sets is the cluster structure[1,32]. In networks, the analogous structure is called communities, which are groups of densely connected nodes that have few connections between groups (also called partitions)[33]. The nodes in chronnet that correspond to an active region (multiple events) tend to be connected forming a community. If the region of activity changes to another region, another community emerges. Thus, communities in chronnets represent groups of cells (regions) whose events appear consecutively in an interval.

**Spatiotemporal data mining.** The chronnet construction method itself is a form of detecting frequent patterns. The method concentrates on a pattern: consecutive events between different locations. This pattern is represented by strong links that indicate repeating consecutive events between different regions. Since we consider different regions, it can also be used for spatiotemporal relationship mining. Strong relationships in terms of consecutive events can be studied by pruning low-weight links or focusing on the strongest links in the chronnet.

In chronnet, cells with long periods of activity are mapped to high degree nodes, and cells with a high number of events are mapped to high strength nodes. These nodes represent cells whose activity is much higher than the other ones and can,

therefore, be treated as outliers. The degree and/or strength can be used to detect these outlier nodes that may be removed from the network when convenient.

In data sets with spatiotemporal clusters, the events in a cluster tend to generate strong links between the nodes representing the geometric region where the events occur, resulting in community structures in chronnets. Different community detection methods have been proposed to find a single partition or a hierarchical clustering structure, commonly represented as a dendrogram[24,33]. Hierarchical community structures are particularly interesting since they permit to define an arbitrary number of communities. Other advantages are the low time complexity and the reduced or absent number of algorithm parameters, making them suitable for large data sets.

After detecting communities, they can be used to find spatiotemporal clusters in the data set. Given a data set $X = \{e_1, e_2, \ldots, e_T\}$ composed by $T$ events, the clustering for each event is defined as the community where the node that represents this event is inserted. This process results in a temporal sequence of communities $C = \{c_1, c_2, \ldots, c_T\}$ defining the spatiotemporal clustering for events. Hierarchical community detection algorithms can be used to divide the nodes into an arbitrary number of communities, making possible the spatial division into a distinct number of regions. An advantage of community detection algorithm over traditional clustering algorithms comes from their partition strategy that normally considers macro, mesoscale, and microcharacteristics from the network[33,34]. This strategy tends to find clusters of different forms and sizes, which might provide better clustering results.

Outliers can be treated by applying a correction function $f(C, \delta)$ to each element $c_t \in C$ where $\delta \in 2\mathbb{N} + 1$ is an odd natural number defining a time window size. This function assigns to every element $c_t$ the same value in the time window centered in $c_t$ iff all the values in the window are the same, i.e., if the number of unique elements in $\{c_{t-\delta}, \ldots, c_{t-1}, c_{t+1}, \ldots, c_{t+\delta}\}$ is one.

The community structure can also be used to find temporal changes in spatiotemporal data sets composed by spatial clusters. Temporal changes can be detected by tracking alterations in the communities where the events occur in time. Following the same approach described for the clustering, we consider a data set $X = \{e_1, e_2, \ldots, e_T\}$ composed of $T$ events and the respective time series of communities $C = \{c_1, c_2, \ldots, c_T\}$ that represent the node community where each event occurs. The time intervals in the time series $C$ where the events persist in a specific community correspond to a particular region with a considerable high number of events. When the community changes, it represents a time point where the events start to appear in another region, indicating spatial changes. Therefore, a time index $t \in \{2, 3 \ldots, T\}$ in the data set $X$ is considered a change point if $c_t \neq c_{t-1}$.

**Experimental settings.** To analyze and test our method, we use three kinds of data: an artificial data set generator, dynamical systems, and a real-world data application. The artificial data sets were constructed with specific characteristics to show the potential of the method and the real data set is a case study where we show different known phenomena via the proposed model. The three approaches are described in the following.

We propose a simple spatiotemporal data set generator composed by an square grid $Gr_{x \times y}$ with $x$ columns and $y$ rows. The probability matrix $P_{x \times y}$, which has the same size of the grid Gr, defines the likelihood $P_{ij}$ of a cell $Gr_{ij}$ generates an event in a time $t \in [1, 2, \cdots, T]$, where $T \in \mathbb{N}$ is an arbitrary time length. Different data sets can be generated by modifying the three parameters: grid size $Gr_{x \times y}$, the probability matrix $P$, and the time length $T$. Figure 9 exemplifies the data generation process.

We also used data extracted from the Lorenz and Rössler equations using well-known combinations of parameters that generate chaotic trajectories in both equations[35]. The data set was constructed by sampling the values in the trajectories considering a fixed time step.

We use a global fire detection data set constructed with data from the moderate resolution imaging spectroradiometer (MODIS) that runs on-board NASA's Terra and Aqua satellites[36]. Specifically, we used the Global Fire Location Product (MCD14ML) Collection 6 from 2003 to 2018[37]. This data set is composed of global

active fire detection with geographic location, date, confidence, type, and additional information for each fire detected by MODIS sensors. We consider only those fire records with confidence higher than 75%. We opt for dividing the globe with a hexagonal grid due to its advantages over a traditional rectangular longitude–latitude one[38]. Hexagonal grids generate cells with a more uniform coverage area and avoid distortions. In our experiments, we used 21,872 hexagonal grid cells of approximated 23,322 km$^2$ each to cover the whole globe. Since many cells cover regions without fire, e.g. the oceans or poles, we discard them resulting in 5467 cells with at least one fire detection in the historical period. The "type" column in the data set defines if the fire detection is a vegetation fire or an outlier generated by active volcanos, offshores, or other static land sources. In our experiments, we used all the types to test whether our method can detect these outlier activities.

## Data availability

The artificial datasets generated and analyzed during the current study can be replicated using our source code that is free for download and use. The MODIS data that support the findings of this study are available at http://modis-fire.umd.edu/files/MODIS_C6_Fire_User_Guide_B.pdf.

## Code availability

Our source code was implemented in R and is free for download at https://github.com/lnferreira/chronnets with the identifier https://doi.org/10.5281/zenodo.3906599. It includes routines to generate artificial data sets and chronnets, as well as other analysis functions.

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

## Acknowledgements

This research is supported by the Fundação de Amparo à Pesquisa do Estado de São Paulo (FAPESP) under Grant No. 2015/50122-0 and the German Research Council (DFG-GRTK) Grant No. 1740/2. L.N.F. acknowledges FAPESP Grant No. 2019/00157-3 and 2017/05831-9. D.A.V.-O. acknowledges FAPESP Grants 2016/23698-1, 2018/24260-5, and 2019/26283-5. M.G.Q. acknowledges FAPESP Grant 2016/16291-2 and CNPq Grant 313426/2018-0. L.Z. acknowledges the Pro-Rectory of Research (PRP) of University of Sao Paulo Grant 2018.1.1702.59.8 and CNPq Grant 303199/2019-9. M.F.C. acknowledges CNPq Grant 314016/2009-0. This research was developed using computational resources from the Center for Mathematical Sciences Applied to Industry (CeMEAI) funded by FAPESP (grant 2013/07375-0).

## Author contributions

L.N.F., D.A.V.O., M.C., M.F.C., and M.G.Q. developed the chronnet model. L.N.F. proposed the data set generator, conceived, and conducted the experiments. L.N.F. and D.A.V.O. analyzed the results and wrote the paper. L.Z. and E.E.N.M. contributed with ideas to improve the work. All authors reviewed the manuscript.

## Competing interests

The authors declare no competing interests.
