## [Peer Review File · Nature Communications]

Reviewers' comments:

Reviewer #1 (Remarks to the Author):

The paper proposes a network based architecture to analyse spatio-temporal data. The method starts by dividing the spatial region under study into grid cells which are represented by nodes in the network. The method aims to capture consecutive recurrent events by modelling them as connections between nodes in the network. The paper builds on existing prior previous work "From Spatio-temporal data to chronological networks: An application to wildfire analysis" from the corresponding author which is mentioned in the introduction. The authors extend their previous work by making it more generalisable to different domains. Moreover, this method of representing spatio-temporal events in the form a network is claimed to be a better way to deal with data mining problems namely, clustering, predictive learning, pattern mining, anomaly detection, change detection and relationship learning.

The experiments are conducted on two datasets: 1) artificially generated spatio-temporal dataset 2) real world global fire detection dataset.

However, the claims in the paper need to be validated through rigorous experiments that ideally go beyond synthetic datasets and wild-fire analyses that was already presented in their previous work. Even for experiments done on the synthetic dataset, this reviewer couldn't find experiments supporting the capability of the network in the context of predictive learning, pattern mining and change detection

The paper has many presentation issues. The grammar and typos needs to be thoroughly checked, and references should be consistent. For example, many references have all the authors listed while several of them have the authors in et al. format.

Farkas, I., Jeong, H., Vicsek, T., Barabási, A.-L. & Oltvai, Z. The topology of the transcription regulatory network in the yeast, *Saccharomyces cerevisiae*. *Phys. A: Stat. Mech. its Appl.* 318, 601–612 (2003).

22. Zou, Y. et al. Brain anomaly networks uncover heterogeneous functional reorganization patterns after stroke. *NeuroImage: Clin.* 20 (2018).

Boers, N. et al. Complex networks reveal global pattern of extreme-rainfall teleconnections. *Nature* 566, 373 (2019).

Reviewer #2 (Remarks to the Author):

The authors present a new tool, called chronnet, to extract important information out of data from spatio-temporal systems. They first construct a network from 2D-data and analyze then data from model as well as observed fires by means of several networks characteristics.

This is an important problem of high actuality. The paper is well written; the methods and the results are well explained.

But I have several questions which should be considered before I can make a final recommendation:

- It should be pointed out that the method is restricted to 2D data, but often one has 3D data in

nature. Is it possible to extend it to 3D?

- The definition of an "event" is also not trivial and could lead to several artifacts.
- There could be (almost) parallel events at different spatial locations. How to treat them?
- Self-connections are allowed here. But they are typically excluded in networks.
- The pruning parameter seems to be somewhat artificial and could also lead to artifacts.
- There are several techniques in the complex network literature to identify the most influential nodes. They should be used for comparison here at least and may be applied.
- What do you really learn new about fires with your method?

Reviewer 1

We would like to thank reviewer #1 for his/her time, for carefully reviewing our article, and for the considerations that have surely improved the manuscript.

the claims in the paper need to be validated through rigorous experiments that ideally go beyond synthetic datasets and wild-fire analyses that was already presented in their previous work. Even for experiments done on the synthetic dataset, this reviewer couldn't find experiments supporting the capability of the network in the context of predictive learning, pattern mining and change detection

We agree with the reviewer that additional experiments can better elucidate the method's capability. Therefore, in the revised version, we have added experimental analyses on four new spatiotemporal data sets generated by different models, including the Lorenz and Rössler models. The new study provides more evidence that the Chronnets can properly capture the spatiotemporal patterns of the data sets under study, for example, the double-wing structure of the chaotic Lorenz attractor (see Figure 5 and the corresponding explanation).

Still following the reviewer's suggestion, we added to the manuscript another experiment to support the method's capability in a new spatiotemporal data mining task: change detection. We created an artificial data set using the data generator proposed in the paper composed of repeating consecutive events between cells, a behavior that is captured by our model. Spatiotemporal changes in the data set can be observed by tracking the temporal community where the events occur. The experimental result is illustrated in the new Fig. 7. We hope that this new experiment makes clear to the reviewer that our method can be used in almost all the spatiotemporal data mining tasks, showing itself a prominent tool.

Our experimental results focused on presenting evidence that this model captures temporal phenomena previously observed in artificial and real data sets. We worked with both artificial data sets and wildfire data set that were not used before. The artificial data sets were proposed in this paper, and the results have not been published in our previous work [5]. The same happens to the real data set. The wildfire data set used in the previous work focus in a specific region (the Amazon basin), and here we use global data, which changes

the analysis and results completely. The larger area permits the method to capture not just the fire seasons in a specific region but also the dynamics on a global scale. The difference in data set sizes also demonstrates the method’s capability of dealing with huge data sets. Thus, we reaffirm that all the results presented in this paper were not previously published.

The main goal of this paper is to present a network-based method that captures frequent consecutive events. Another goal is to make possible the use of network science and graph mining tools to extract information from spatiotemporal data. The research areas mentioned by the reviewer involves different tasks with many works in the literature [2]. In our paper, we selected four tasks [1]: frequent pattern mining, relationship mining, outlier detection, and clustering. For these four tasks, we experimentally demonstrate how to use the model together with other graph mining tools to detect temporal patterns. However, it is relevant to mention that our method is not limited to the analysis tools described in the manuscript. Virtually all previous works [6] in the context of predictive learning, pattern mining, and change detection in networks can be applied to this model, which is an advantage. For example, in the context of predictive learning, many methods for link prediction in networks [4] have been developed and can be applied or adapted to our model. Given the complexity of these tasks, we believe that exploring them in this paper would diverge from our focus, which is the chronnet model proposal. The application of other graph mining tools to our model opens many possibilities for future works. We are certain that they will be used together to extract information from different spatiotemporal data sets.

The paper has many presentation issues. The grammar and typos needs to be thoroughly checked, and references should be consistent. For example, many references have all the authors listed while several of them have the authors in et al. format.

We reviewed the whole manuscript and tried our best to remove typos. We also correct the references. Please note that we used the recommended latex bibliographical style for the *Nature journals*, which automatically shrinks the references to (et al.) according to the number of authors.

Reviewer 2

We would like to extend our gratitude to reviewer #2 for his/her time, for carefully reviewing our article, and for the considerations that have certainly improved our paper.

- It should be pointed out that the method is restricted to 2D data, but often one has 3D data in nature. Is it possible to extend it to 3D?

In this paper, we focus on spatiotemporal events, commonly defined as triples: (x,y,t) representing the location and the time in which observation occurs. So our method, as presented in the paper, takes into account three dimensions (not only 2D). For simplicity reasons, we opt for focusing on the three dimensions because they are the most simple and common way to represent spatiotemporal events. In our method, x and y are divided into grid cells represented by nodes, and the temporal variable t defines the order where the links are

established. If the spatial grid is, for example, 10×10 , it will lead to a chronnet of 100 nodes. This grid transformation is simply a binning operation in the variables x and y to find the cell (and the node) where an event occurs. One possible way to extend it to one variable z (or more variables) consists of applying the same binning operation for the grid to this new variable. Considering now a 4D data set (x,y,z,t) and a grid $10 \times 10 \times 10$, it will lead to a chronnet of 1000 nodes. In summary, more variables can be considered in the chronnet model by increasing the number of dimensions of the grid. Then, our method is not restricted to three dimensions, but it can process higher-dimensional data. We included this explanation in the manuscript.

- The definition of an "event" is also not trivial and could lead to several artifacts.

We define an event as a spatiotemporal observation, i.e., a record in a spatiotemporal data set. Since these observations are application-dependent, they represent different occurrences in a different data set, but all events have a common characteristic: spatial and temporal values that define where and when they occur. Please note that this definition is widely used in spatiotemporal data analysis [1]. What our method does is transforming a spatiotemporal data set into a network, so it can be used as a general model to study spatiotemporal data set. We do not introduce nor create any record in the data set. If the data set contains incorrect records, this is a data collection issue that can be solved during the data set construction or minimized in the pre-processing step. However, this is not a problem with the method itself.

- There could be (almost) parallel events at different spatial locations. How to treat them?

This is an important question and we included an explanation for this question in the revised version of the manuscript. For simplicity reason and without loose of generality, we consider in the manuscript that time is discrete $\{t_1, t_2 \dots \leq t_T\}$ and the time window is $h = 1$. Let $\{v_1, v_2, \dots, v_r\}^t$ and $\{v_1, v_2, \dots, v_s\}^{t+1}$ be the sets of r and s vertices whose cells have simultaneous events in time t and $t+1$ respectively. In this case, a link is established between all the combinations of different vertices between both sets.

- Self-connections are allowed here. But they are typically excluded in networks.

Self-connections represent consecutive events that occur in the same cell. This is additional information captured in the model that might be used or not. If this information is not relevant for the user, self-connections can be easily excluded. Please, note that typical network transformations and simplifications (e.g., node, self-links, or direction removal) may be applied if convenient or necessary. The model enables the use of any network science or graph mining tool to extract information from the spatiotemporal data set, and this is an advantage of the model.

- The pruning parameter seems to be somewhat artificial and could also lead to artifacts.

The pruning process works as a filter that removes weak links. This process does not introduce data or transform them. It only excludes consecutive events that are not relevant or do not occur sufficiently frequently and can be unconsidered. This procedure is another

tool for the model that permits the user to focus on consecutive events that repeat more frequently (stronger links). This parameter depends on the application and might be used or not according to the explored data set. The pruning procedure can also be interpreted as an outlier removal process, making the method more robust to wrong measurements in the modeled data set. We added to the method description section an explanation about the pruning process.

- *There are several techniques in the complex network literature to identify the most influential nodes. They should be used for comparison here at least and may be applied.*

Following the reviewer’s recommendation, we added in the revised version examples of how to apply centrality measures in the context of chronnets. It is important to mention that many centrality measures do not take into account the link weights, which is essential information for the model. We do not recommend merely disregarding the weights because, without them, the centrality measures would consider that all links have the same influence. Therefore, we suggest two approaches: (1) to use centrality measures for weighted graphs or (2) to prune low weight (spurious) links and then apply centrality measures for unweighted networks. We added in the method section a description of how to apply centrality measures, and we added the new Fig. 5, which presents applications of well-known centrality measures in pruned or considering the weights in chronnets. We show that these measures can find influential nodes in chronnets. Centrality measures are one of the many network science tools that can be applied to our model. We discuss in this paper only the most common network measures, but as we mentioned before, this model opens many possibilities to be explored in future works.

- *What do you really learn new about fires with your method?*

We thank the reviewer for raising this question. As we informed in the experimental settings, we use this data set to observe, via the proposed method, features that we already know. We apply our method to a real-world data set to present experimental evidence that it can extract spatiotemporal information that was previously observed in the literature. For example, the outlier events (“type” column) in the MCD14ML data set [3] was used to test if they correspond to the expected high degree nodes in the chronnet (Fig. 8). Furthermore, we describe the frequency of fire events, outlier fire detections, and the seasonal activity, using a single chronnet. We have chosen this data set as a prove-of-concept of our analysis: it has temporal patterns and is considerably large. However, it could have been any other event-based spatiotemporal data set, which is an advantage of our method. This method opens the possibility to analyze other applications and spatiotemporal domains, which we will consider in future works.

References

- [1] Gowtham Atluri, Anuj Karpatne, and Vipin Kumar. Spatio-temporal data mining: A survey of problems and methods. *ACM Comput. Surv.*, 51(4):83:1–83:41, August 2018.

- [2] D.J. Cook and L.B. Holder. *Mining Graph Data*. Wiley, 2006.
- [3] Louis Giglio. MODIS Collection 6 Active Fire Product User’s Guide Revision A. http://modis-fire.umd.edu/files/MODIS_C6_Fire_User_Guide_A.pdf, 2015. [Online; accessed 01-March-2019].
- [4] Linyuan Lü and Tao Zhou. Link prediction in complex networks: A survey. *Physica A: Statistical Mechanics and its Applications*, 390(6):1150 – 1170, 2011.
- [5] Didier A. Vega-Oliveros, Moshé Cotacallapa, Leonardo N. Ferreira, Marcos G. Quiles, Liang Zhao, Elbert E. N. Macau, and Manoel F. Cardoso. From spatio-temporal data to chronological networks: An application to wildfire analysis. In *34th ACM/SIGAPP Symposium on Applied Computing, SAC '19*, pages 675–682, New York, NY, USA, 2019. ACM.
- [6] M. Zanin, D. Papo, P.A. Sousa, E. Menasalvas, A. Nicchi, E. Kubik, and S. Boccaletti. Combining complex networks and data mining: Why and how. *Physics Reports*, 635:1 – 44, 2016.

REVIEWERS' COMMENTS:

Reviewer #1 (Remarks to the Author):

Thanks for taking the time to respond to my comments and revising the manuscript accordingly. The revised version is much improved and could be accepted for publication. However I still like to express my concern that this paper (even in its revised form) relies too much on synthetic data sets to illustrate the utility of the proposed method. A reader's trust in the method will be greatly strengthened if the authors have attempted to evaluate it on several real life data sets.

Reviewer #2 (Remarks to the Author):

The authors answered comprehensively to all my points in this revised ms. Therefore, I can recommend it now for publication.